# Sesamol Alleviates Airway Hyperresponsiveness and Oxidative Stress in Asthmatic Mice

**DOI:** 10.3390/antiox9040295

**Published:** 2020-04-01

**Authors:** Chian-Jiun Liou, Ya-Ling Chen, Ming-Chin Yu, Kuo-Wei Yeh, Szu-Chuan Shen, Wen-Chung Huang

**Affiliations:** 1Division of Basic Medical Sciences, Department of Nursing, Research Center for Chinese Herbal Medicine, and Graduate Institute of Health Industry Technology, Chang Gung University of Science and Technology, No.261, Wenhua 1st Rd., Guishan Dist., Taoyuan City 33303, Taiwan; ccliu@mail.cgust.edu.tw; 2Division of Allergy, Asthma, and Rheumatology, Department of Pediatrics, Chang Gung Memorial Hospital, Linkou, Guishan Dist., Taoyuan City 33305, Taiwan; kwyeh@cgmh.org.tw; 3School of Nutrition and Health Sciences, Taipei Medical University, 250 Wu-Hsing Street, Taipei City 11031, Taiwan; ylchen01@tmu.edu.tw; 4Department of Surgery, Chang Gung Memorial Hospital, Linkou, Guishan Dist., Taoyuan City 33305, Taiwan; mingchin2000@gmail.com; 5Graduate Program of Nutrition Science, National Taiwan Normal University, 88 Ting-Chow Rd., Sec 4, Taipei 11676, Taiwan; 6Graduate Institute of Health Industry Technology, Research Center for Food and Cosmetic Safety, Research Center for Chinese Herbal Medicine, College of Human Ecology, Chang Gung University of Science and Technology, No.261, Wenhua 1st Rd., Guishan Dist., Taoyuan City 33303, Taiwan

**Keywords:** airway hyperresponsiveness, asthma, eosinophil infiltration, oxidative stress, sesamol

## Abstract

Sesamol, isolated from sesame seeds *(Sesamum indicum)*, was previously shown to have antioxidative, anti-inflammatory, and anti-tumor effects. Sesamol also inhibited lipopolysaccharide (LPS)-induced pulmonary inflammatory response in rats. However, it remains unclear how sesamol regulates airway inflammation and oxidative stress in asthmatic mice. This study aimed to investigate the efficacy of sesamol on oxidative stress and airway inflammation in asthmatic mice and tracheal epithelial cells. BALB/c mice were sensitized with ovalbumin, and received oral sesamol on days 14 to 27. Furthermore, BEAS-2B human bronchial epithelial cells were treated with sesamol to investigate inflammatory cytokine levels and oxidative responses in vitro. Our results demonstrated that oral sesamol administration significantly suppressed eosinophil infiltration in the lung, airway hyperresponsiveness, and T helper 2 cell-associated (Th2) cytokine expressions in bronchoalveolar lavage fluid and the lungs. Sesamol also significantly increased glutathione expression and reduced malondialdehyde levels in the lungs of asthmatic mice. We also found that sesamol significantly reduced proinflammatory cytokine levels and eotaxin in inflammatory BEAS-2B cells. Moreover, sesamol alleviated reactive oxygen species formation, and suppressed intercellular cell adhesion molecule-1 (ICAM-1) expression, which reduced monocyte cell adherence. We demonstrated that sesamol showed potential as a therapeutic agent for improving asthma.

## 1. Introduction

Asthma is an allergic respiratory disease that is important globally. The pathological characteristics of allergic asthma include inflammation and allergic reactions in the airways and increased eosinophil infiltration in the lungs. These conditions lead to airway hyperresponsiveness (AHR), airway remodeling due to smooth muscle hyperplasia, and narrow airways [1]. Sudden asthma attacks are characterized by paroxysmal wheezing, dry cough, shortness of breath, and chest tightness [2]. In patients with asthma, airway smooth muscle contraction and goblet cell hyperplasia stimulate mucus secretion, which obstructs the airway and causes difficulty breathing [3]. Hence, severe asthma attacks require urgent medication to avoid suffocation.

The pathological development of asthma is mainly related to excessive activation of T helper 2 (Th2) lymphocytes. Activated Th2 cells can secrete excessive cytokines, including interleukin (IL)-4, IL-5, and IL-13, which affect AHR, airway remodeling, eosinophil infiltration, and excessive mucus secretion in the trachea [4]. In recent years, clinical drugs have been developed to improve the symptoms of asthma with potential therapeutic effects on IL-4, IL-5, and IL-13 expression levels [5]. However, some patients with asthma have not responded well to these new treatments. Therefore, it is necessary to develop and design other treatment methods or drugs to regulate the effects of Th2 activation in patients with asthma.

Tracheal epithelial cells can defend against microorganism and allergen invasions to reduce the entry of harmful substances into the lungs. Allergens and inflammatory mediators stimulate tracheal epithelial cell activation [6]. Activated tracheal epithelial cells release cytokines, which induce an inflammatory response, and chemokines, which attract more immune cells to the lungs. These immune cells release more inflammatory and oxidative molecules to increase lung inflammation, which damages lung cells and tissues [7]. In addition, inflammatory epithelial cells can stimulate the expression of oxidases, such as nicotinamide adenine dinucleotide phosphate (NADPH) oxidase and inducible nitric oxide synthase, which produce reactive oxygen species (ROS) and nitric oxide. These conditions cause oxidative stress, which leads to airway remodeling, smooth muscle thickening, and pulmonary cell damage [8]. Therefore, in the airways, ROS and inflammatory response cytokines are dangerous signaling molecules that cause persistent respiratory cell damage and respiratory dysfunction. 

*Sesamun indicum*, Linn. (Sesame) is a common source of sesame seeds, which can be refined to produce sesame oil. This edible vegetable oil contains high levels unsaturated fatty acids, including linolenic acid and linoleic acid [9]. Sesamol is a polyphenol lignan isolated from sesame seeds. Sesamol has shown multiple biologically active functions in animal and cellular disease models [10,11]. Sesamol alleviated the expression of inflammatory mediators by suppressing the nuclear factor-κB (NF-κB) and mitogen-activated protein kinase (MAPK) signaling pathways. Sesamol also displayed antioxidant effects by promoting nuclear factor erythroid 2-related factor 2/ heme oxygenase-1 (Nrf2/HO-1) expression in LPS-stimulated macrophages [12]. Moreover, sesamol reduced oxidative stress by regulating the sirtuin 1/ forkhead box class O 3a (SIRT1/FOXO3a) signaling pathway in human neuronal cells [13]. Recently, sesamol was shown to improve neutrophil infiltration and the inflammatory response in mouse lungs after an LPS-induced acute lung injury [14]. However, it remains unclear whether sesamol might improve airway inflammation, eosinophil infiltration, and oxidative stress in asthmatic lungs. In the present study, we investigated whether sesamol could regulate the molecular mechanisms involved in oxidative stress and the inflammatory response in an ovalbumin (OVA)-induced asthmatic mouse model and in inflamed human tracheal epithelial cells.

## 2. Materials and Methods 

### 2.1. Animals

Six-week-old female BALB/c mice were purchased from the National Laboratory Animal Center, Taiwan. All mice were raised in air-conditioned animal housing under a 12-h light/dark cycle with food and water ad libitum. All animal experiment protocols in this study were approved by the Laboratory Animal Care Committee of Chang Gung University of Science and Technology (IACUC approval number: 2018-003).

Sesamol (≥98% purity; Sigma-Aldrich, St. Louis, MO, USA) was dissolved in normal saline before administration. Mice were divided into the following five treatment groups (each group, n = 8): (1) Mice were sensitized with normal saline and received normal saline, administered orally (N group); (2) mice were sensitized with OVA and received normal saline orally (OVA group); (3) mice were sensitized with OVA and were fed 10 or 30 mg/kg sesamol (S10 and S30 groups, respectively); and (4) mice were sensitized with OVA and were fed 10 mg/kg prednisolone (P group; the positive control).

### 2.2. Establishment of an Asthma Model and Sesamol Administration

Asthma was induced in mice as shown in Figure 1A. Briefly, mice were sensitized with intraperitoneal injections of a sensitization solution (0.8 mg aluminum hydroxide; Thermo, Rockford, IL) and 50 μg ovalbumin (OVA; Sigma) in 200 μL of normal saline) on days 1-3 and 14. Next, mice inhaled 2% OVA administered with an ultrasonic nebulizer for 30 min to induce asthma symptoms (DeVilbiss Pulmo-Aide 5650D, United States) on days 14, 17, 20, 23, and 27. Then, for 2 weeks (days 14-27), in addition to their regular diet, each mouse group received a daily dose of saline (N and OVA groups), sesamol (S10 and S30 groups), or prednisolone (P group). On day 28, we calculated the AHR in all mice; then, on day 29, we sacrificed mice to investigate the asthma pathology, immune regulation, inflammation, and oxidative stress. 

### 2.3. Airway Hyperresponsiveness Assay

Airway function was examined after administering aerosolized methacholine, as described previously [15]. Briefly, methacholine doses were prepared (0, 10, 20, 30, and 40 mg/mL) in normal saline solution, and mice inhaled various doses of aerosolized methacholine for 3 min. Then, mice were placed in a closed system to record the enhanced pause (Penh) with a whole-body plethysmograph; the Penh was used to calculate the AHR (Buxco Electronics, Troy, NY, USA) on day 28.

### 2.4. Histological Analysis of Lung Tissue

Mice were anesthetized and sacrificed. Lung tissues were removed and fixed in 10% neutral buffered formalin (Sigma). Lung tissues were embedded into paraffin and cut into 6-μm-thick sections. Then, the lung tissue sections were stained with a hematoxylin and eosin (HE) solution to calculate eosinophil infiltration (pathological scores), based on a five-point grading system described previously [16]. Briefly, the pathological degree of eosinophil infiltration was scored as follows: 0, no cell; 1, a few cells; 2, a ring of inflammatory cells one cell layer; 3, a ring of inflammatory cells two to four cells layer; and 4, a ring of inflammatory cells > four cells layer [17]. Lung sections were also stained with a periodic acid-Schiff (PAS) solution (Sigma) to evaluate goblet cell hyperplasia in the trachea and bronchial epithelium, as described previously [17]. 

### 2.5. Bronchoalveolar Lavage Fluid and Cell Counting

Mice were anesthetized and sacrificed to collect bronchoalveolar lavage fluid (BALF), as previously described [18]. Briefly, an indwelling needle was inserted into the trachea, and the end was connected to a 1-mL syringe. The lungs and airways were washed by loading lavage fluid into the syringe, then moving the plunger back and forth three times. Next, the BALF was centrifuged at 1500 rpm for 5 min, and the supernatant was collected and stored at −80 °C. These samples were later assayed to detect cytokine and chemokine levels. Furthermore, BALF cells were treated with red blood cell lysis buffer and stained with Giemsa stain (Sigma). Then, cells were counted to determine the total cell number and the numbers of different types of immune cells.

### 2.6. Serum Collection

Mice were anesthetized and blood was collected from a submandibular vein. Blood was centrifuged to obtain the serum. Serum was analyzed with an ELISA to detect OVA-specific antibodies, as previously described [19]. 

### 2.7. ELISA Assay

We analyzed the supernatants of cell cultures and BALF from mice with commercial ELISA kits (R&D Systems, Minneapolis, MN), according to the manufacturer’s instructions. We determined the levels of IL-6, tumor necrosis factor-α (TNF-α), IL-4, IL-5, IL-8, IL-13, intercellular adhesion molecule 1 (ICAM-1), monocyte chemoattractant protein-1(MCP-1), chemokine (C-C motif) ligand 24 (CCL24), CCL11, and CCL5, as previously described [17]. Furthermore, we used a specific ELISA kit (BD Biosciences, San Diego, CA) to determine serum OVA-specific immunoglobulin (Ig) E, IgG1, and IgG2a levels. For OVA-IgE levels, serum samples were diluted 5-fold and the optical density (OD) was measured at 450 nm. OVA-IgG1 and OVA-IgG2a standard curves were created with mixed serum samples from OVA-sensitized mice. Cytokine and antibody levels were detected at an OD of 450 nm with a microplate reader (Multiskan FC, Thermo, Waltham, MA, USA).

### 2.8. Malondialdehyde Activity 

Malondialdehyde (MDA) activity in the lung was examined with a lipid peroxidation assay kit (Sigma), as described previously [17]. Briefly, lung tissues were homogenized and centrifuged, and the supernatants were isolated. Then, we added perchloric acid to the supernatants to precipitate the proteins. These samples were centrifuged, and the supernatant was collected. We detected MDA activity with a multi-mode microplate reader (SpectraMax i3X, Molecular Devices, San Jose, CA, USA). 

### 2.9. Glutathione Assay

The glutathione (GSH) levels in lung tissues were examined with a glutathione assay kit (Sigma), as described previously [17]. Briefly, lung tissues were homogenized in 5% 5-sulfosalicylic acid in a homogenizer (FastPrep-24, MP Biomedicals, Santa Ana, CA, USA). We centrifuged the samples and collected the supernatant. GSH levels were detected with a microplate reader (Thermo) at an OD of 412 nm. 

### 2.10. Real-Time PCR Analysis 

Total RNA was extracted with TRI reagent (Sigma). cDNA was reverse transcribed from 100 ng RNA with a cDNA synthesis kit (Bio-Rad, San Francisco, CA, USA), according to the manufacturer’s instructions. Next, specific genes were PCR-amplified and quantified with the SYBR Green Master Mix kit (Bio-Rad) and a spectrofluorometric thermal cycler (iCycler; Bio-Rad), as described previously [15]. The PCR conditions were: 95 °C for 10 min, and 40 cycles of 95 °C for 15 s and 60 °C for 60 s. 

### 2.11. BEAS-2B Cell Culture and Sesamol Treatment

Sesamol was dissolved in phosphate buffered saline (PBS), and a 100 mM stock solution was prepared. Human bronchial epithelial cells (BEAS-2B, American Type Culture Collection, Manassas, VA) were cultured in dulbecco’s modified eagle medium (DMEM)/F12 medium, and seeded into 24-well plates. Cell viability was determined using the 3-(4,5-Dimethylthiazol-2-yl)-2,5- diphenyltetrazolium bromide) (MTT) solution (Sigma), as described previously [20]. BEAS-2B cells were treated with sesamol (0–100 μM) for 1 h, then cells were stimulated with or without 10 ng/mL TNF-α and 10 ng/mL IL-4 for 24 h. Harvested cells were centrifuged, and the supernatants were collected. We determined chemokine and cytokine production with specific commercial ELISA kits.

### 2.12. Cell-Cell Adhesion Assay

BEAS-2B cells were treated with various concentrations of sesamol and stimulated with or without TNF-α/IL-4 for 24 h. Human monocytic cells (THP-1, Bioresource Collection and Research Center, Taiwan) were cultured in RPMI 1640 medium. THP-1 cells were stained with calcein-AM solution (Sigma). Next, THP-1 cells were co-cultured with BEAS-2B cells, and adherent THP-1 cells were evaluated with fluorescence microscopy (Olympus, Tokyo, Japan), as previously described [21].

### 2.13. Reactive Oxygen Species Assay

BEAS-2B cells were treated with sesamol for 1 h, and stimulated with or without 10ng/mL TNF-α/IL-4 for 24 h. Then, BEAS-2B cells were incubated with 20 μM dichloro-dihydro-fluorescein diacetate for 30 min, as previously described [17]. Briefly, three images were calculated for each result, and three areas were selected for each image to quantify the fluorescence intensity with a fluorescence microscope (Olympus). Cells were then lysed, and ROS levels were measured by exciting at 485 nm and recording emissions at 528 nm with a Multi-Mode microplate reader (SpectraMax i3X, Molecular Devices). 

### 2.14. Statistical Analysis

Statistical analyses were performed with a one-way analysis of variance (ANOVA), followed by Dunnett’s post hoc test for normally distributed data. Non-normally distributed data used non-parametric Kruskal–Wallis analysis. All data are expressed as the means ± standard error of the mean (SEM), based on at least three independent experiments. *p*-values <0.05 were considered significant.

## 3. Results

### 3.1. Sesamol Effects on AHR in Asthmatic Mice

Asthmatic mice received oral sesamol administrations once daily from days 14 to 27. On day 28, all mice inhaled increasing doses (0–40 mg/mL) of methacholine to examine AHR. We found that the Penh values were significantly higher in the OVA group of asthmatic mice compared to normal mice (Figure 1B). At 40 mg/mL of inhaled methacholine, we observed significant attenuations in the Penh values of the sesamol and prednisolone groups, compared to the OVA group. Thus, oral sesamol administration inhibited AHR in asthmatic mice.

### 3.2. Effect of Sesamol on Inflammatory Cells in BALF

In BALF, inflammatory cells were stained with Giemsa. We found significantly higher numbers of eosinophils in the OVA group of asthmatic mice (7.1 × 10^5^ ± 7.8 × 10^4^) compared to normal mice. Moreover, the numbers of eosinophils in OVA-treated asthmatic mice were significantly reduced with a high sesamol concentration (S10: 5.7 × 10^5^ ± 5.8 × 10^4^, *p* = 0.21; S30: 4.6 × 10^5^ ± 6.3 × 10^4^, *p* < 0.05) and prednisolone (3.5 × 10^5^ ± 4.4 × 10^4^, *p* < 0.01). The total cell numbers in asthmatic mice were significantly reduced after treatment with sesamol (S10: 1.29 × 10^6^ ± 1.27 × 10^5^, *p* = 0.06; S30: 1.09 × 10^6^ ± 9.11 × 10^4^, *p* < 0.05) or prednisolone (8.5 × 10^5^ ± 6.91 × 10^4^, *p* < 0.01), compared to the OVA group (1.46 × 10^6^ ± 9.15 × 10^4^; Figure 1C). 

### 3.3. Sesamol Effects on Eosinophil Infiltration and Goblet Cell Hyperplasia in Lungs

HE staining demonstrated that more eosinophils infiltrated the lungs of asthmatic mice, compared to normal mice. In OVA-sensitized mice, sesamol or prednisolone suppressed lung eosinophil infiltration compared to the OVA group of asthmatic mice (Figure 2A). Sesamol also significantly improved the inflammatory pathology score in asthmatic mice (Figure 2B). PAS staining showed goblet cell hyperplasia in the tracheas of OVA-sensitized mice. Both sesamol and prednisolone significantly reduced goblet cell hyperplasia compared to the OVA group (Figure 2C,D).

### 3.4. Sesamol Effects on Cytokine and Chemokine Levels in BALF and Lung Tissue 

In BALF, sesamol significantly reduced the levels of IL-4, IL-5, IL-13, IL-6, TNF-α, CCL11, and CCL24, compared to OVA-sensitized asthmatic mice (Figure 3). Moreover, an analysis of the gene expression in lung tissues showed that sesamol inhibited IL-4, IL-5, IL-13, IL-6, TNF-α, cyclooxygenase (COX)-2, CCL11, and CCL24 expression compared to asthmatic mice (Figure 4). Conversely, sesamol significantly promoted interferon (IFN)-γ levels in BALF and lung tissues, compared to asthmatic mice (Figure 3 and Figure 4).

### 3.5. Sesamol Modulated GSH and MDA Activities in Lung issues

Asthma attacks can induce oxidative stress in the lung and attenuate lung function [22]. We homogenized mouse lung tissues to assay GSH and MDA levels. We found that sesamol significantly enhanced GSH levels and reduced MDA activity compared to OVA-treated asthmatic mice (Figure 5A,B). 

### 3.6. Sesamol Modulated ICAM-1 and Mucin 5AC (Muc5Ac) Expression in Lung Tissues

We performed PCR to detect ICAM-1 and Muc5Ac gene expression in lung tissues. We found that sesamol significantly reduced ICAM-1 expression (Figure 5C) and suppressed Muc5AC expression in asthmatic mice (Figure 5D).

### 3.7. Sesamol Effect on Serum OVA-Specific Antibodies

Sesamol significantly reduced the serum levels of OVA-IgE and OVA-IgG1, compared to OVA-treated mice (Figure 6A,B). Interestingly, sesamol significantly increased the OVA-IgG2a levels, compared to OVA-treated mice (Figure 6C). In contrast, oral prednisolone administration significantly suppressed the levels of all OVA-specific antibodies, compared to OVA-treated mice (Figure 6).

### 3.8. Sesamol Suppressed Proinflammatory Cytokine and Chemokine Production in BEAS-2B Cells 

In this study, the cytotoxicity of sesamol in BEAS-2B cells was determined using the MTT assay. Sesamol did not show significant cytotoxic effects at doses ≤100 μM, and subsequent experiments used sesamol at 10–100 μM concentrations (data not shown). BEAS-2B cells treated with various sesamol concentrations were stimulated with 10 ng/mL TNF-α/IL-4. We found that sesamol significantly dose-dependently reduced the levels of CCL11, CCL24, CCL5, MCP-1, IL-6, and IL-8, compared to BEAS-2B cells stimulated with TNF-α/IL-4 alone (Figure 7). 

### 3.9. Sesamol Suppressed THP-1 Cell Adhesion to BEAS-2B Cells

We found that TNF-α/IL-4 stimulated ICAM-1 expression in BEAS-2B cells, and sesamol significantly reduced ICAM-1 levels after TNF-α/IL-4 treatment (Figure 8A). Next, we stained THP-1 cells with calcein-AM and co-cultured them with BEAS-2B cells. We found that sesamol significantly reduced THP-1 cell adherence to TNF-α/IL-4-treated BEAS-2B cells (Figure 8B,C). 

### 3.10. Sesamol Effects on ROS Production 

BEAS-2B cells were incubated with dichloro-dihydro-fluorescein diacetate to label intracellular ROS. Fluorescence microscopy showed that sesamol reduced the intracellular ROS levels in BEAS-2B cells treated with TNF-α/IL-4 (Figure 9A,B). Furthermore, when ROS levels were measured with a Multi-Mode microplate reader, we found that sesamol significantly attenuated ROS levels, compared to BEAS-2B cells treated with TNF-α/IL-4 (Figure 9C).

## 4. Discussion

Sesamol is a lignin isolated from sesame seeds [7]. Sesamol has many biological functions, which produce anti-inflammatory, antioxidant, and anti-tumor effects [10]. Studies have shown that sesamol could induce apoptosis in cancer cells and inhibit the effects of proliferation and autophagy in lung cancer and colon cancer [11]. We and other research groups also discovered that sesamol could inhibit the secretion of inflammatory cytokines and reduce oxidative damage caused by inflammatory macrophages [12,23]. Other studies found that sesamol could ameliorate the effects of oxidative stress and inflammation in focal cerebral ischemia/reperfusion injury in the rat brain [24]. Sesamol also reduced inflammatory cell secretions of TNF-α and IL-6, promoted superoxide dismutase expression to increase antioxidation effects, and reduced neutrophil infiltration in a rat model of LPS-induced acute lung injury [14]. Taken together, these findings have shown that sesamol is a natural lignin with excellent antioxidant and anti-inflammatory properties. Therefore, we hypothesized that sesamol might improve inflammation and show antioxidant effects in asthmatic lungs. 

Excessive Th2 cell activation is considered the main factor involved in pathological symptoms in the lung during asthma development [25]. In the lungs of patients with asthma, Th2 cells secrete excess IL-4, IL-5, and IL-13 to inhibit the expression of Th1 cells, which reduces IFN-γ secretion [4]. Th2 cells secrete excess IL-4 to induce B cell activation and the secretion of more IgE antibodies [26]. When allergens combine with IgE and mast cells, the complex induces mast cells to release an excess of histamine and leukotriene, which causes a severe allergic reaction [27]. We found that sesamol could inhibit IL-4 expression in BALF and lungs, and it reduced serum OVA-specific IgE levels in asthmatic mice. Many previous studies have shown that Th1 cell-associated cytokines could stimulate B cells to secrete IgG2a, and that Th2 cytokines could induce B cells to produce IgG1 in mice [28]. Prednisone has anti-inflammatory and immunosuppressive properties. Prednisone could suppress Th1 and Th2 cell activity to decrease inflammation and allergic reactions in patients with asthma [5]. Hence, asthmatic mice treated with prednisone could reduce inflammation of the lungs and improve asthma symptoms in mice by blocking the expression of Th1- and Th2-associated cytokines. Interestingly, our results showed that sesamol inhibited OVA-IgG1 expression, promoted OVA-IgG2 levels in serum, and increased INF-γ production in the BALF and lungs of asthmatic mice. Therefore, we demonstrated that oral sesamol administration could increase Th1 cell activity and inhibit Th2 cell activity, which improved the pathological features of asthma.

Asthma is an allergic inflammatory respiratory disease. Patients with chronic asthma continuously display chronic inflammation and oxidative stress [8]. Furthermore, pulmonary epithelial cells and immune cells are continuously activated, and they release various inflammatory cytokines, inflammatory mediators, and oxidative molecules that damage cells and tissues, which reduces lung function [6,29]. Previous studies have shown that some natural products could reduce airway inflammation in asthmatic mice [30,31]. For example, asthmatic mice treated with an intraperitoneal injection of curcumin showed reduced IL-6 and TNF-α secretion in BALF [32]. Intraperitoneal injections of casticin and licochalcone A also improved airway inflammation in asthmatic mice, by suppressing inflammatory cytokines and chemokines in inflammatory tracheal epithelial cells [9,19]. In the present study, sesamol was dissolved in PBS or normal saline, and it was administered orally to asthmatic mice with a gastric tube. We found that sesamol reduced inflammation in the BALF and lungs of asthmatic mice. Many previous studies showed that lung tissues accumulate excessive amounts of the inflammatory cytokines, TNF-α and IL-6, which cause damage to lung cells, and might also induce the development of lung fibrosis [33,34]. In addition, allergens and inflammatory cytokines were shown to stimulate an inflammatory response in tracheal epithelial cells and increase lung tissue damage caused by inflammation [25,35,36]. The present study demonstrated that sesamol could attenuate the inflammatory response in TNF-α/IL-4-stimulated tracheal epithelial cells. Therefore, we concluded that sesamol improved lung inflammation in asthmatic mice, which inhibited asthma development.

Airway remodeling is a structural change of the airways in patients with asthma. Airway remodeling includes airway narrowing and enhanced airway resistance [37]. In patients with chronic asthma, airway smooth muscle thickening and reduced connective tissue elasticity reduces the alveolar surface tension and elasticity, which worsens lung compliance [38,39]. Clinically, AHR is used to detect the airflow and respiratory rate in airways to evaluate pathological characteristics and pulmonary function in patients with asthma [1]. Many previous studies have pointed out that oxidative stress and inflammation exacerbated AHR values, which reflected the exacerbation the deterioration of lung function [15,17,19]. We employed whole-body plethysmography to detect AHRs in asthmatic mice. Our results demonstrated that sesamol could reduce AHR, which reflected improved physiological respiratory function. In patients with asthma, excessive Th2 cell activation increases AHR; indeed, IL-13 is a pleiotropic Th2 cytokine that induces AHR [40]. Moreover, IL-13-deficient asthmatic mice showed significantly less AHR than wild-type asthmatic mice [41]. Our results demonstrated that sesamol could attenuate IL-13 gene expression in the lungs and BALF of asthmatic mice. Hence, we concluded that sesamol improved AHR mainly by reducing IL-13 expression in asthmatic mice.

Patients with acute asthma often show large amounts of eosinophil infiltration in the lungs [42]. Many previous studies found that activated eosinophils released increased amounts eosinophil cationic proteins into cellular vesicles. Eosinophil cationic proteins caused respiratory inflammation and induced damage or apoptosis in lung cells, which decreased lung function [43]. In addition, eosinophils released eosinophil peroxidase, which caused oxidative damage to alveolar cells [2]. Th2 cells also released excessive IL-5, which induced bone marrow cells to differentiate to mature eosinophils [42,44]. In patients with asthma, tracheal epithelial cells release high amounts of eotaxin, which induces eosinophil migration and infiltration into the lungs [45]. Therefore, reducing Th2 cell activation and the inflammatory response of tracheal epithelial cells might regulate eosinophilic infiltration in the lung. We found that sesamol reduced the number of total cells and eosinophils in the BALF of asthmatic mice, compared to asthmatic mice. In addition, sesamol inhibited eosinophil infiltration into the lungs of asthmatic mice. Sesamol also reduced IL-5 expression in the BALF and lung, and it reduced eotaxin expression in the lung and tracheal epithelium. Moreover, sesamol reduced ICAM-1 expression in inflamed airway epithelial cells, which inhibited the adherence of THP-1 immune cells. Therefore, we concluded that sesamol improved the allergic and inflammatory responses in asthmatic mice by reducing IL-5 expression in the lung and by inhibiting eosinophil infiltration into the lung.

Respiratory mucus glycoproteins can protect the lungs from microorganisms or allergens that can invade the lower respiratory tract. Additionally, mucus is an important secretory substance for maintaining respiratory functions [3]. However, during asthma onset, the patient’s airway narrows, and excessive mucus secretion increases airway obstruction, which leads to dyspnea or even suffocation [37]. Mucus is mainly secreted by tracheal epithelial cells. Allergens stimulate tracheal epithelial cells to differentiate into goblet cells and proliferate. This leads to excess mucus secretion in patients with asthma [46]. We used the PAS stain to detect goblet cells in asthmatic mice. We found that sesamol reduced goblet cell hyperplasia in the trachea and attenuated Muc5Ac gene expression in the lungs, which reduced mucus production. Previous studies demonstrated that IL-4 and IL-13 were released by overactivated Th2 cells and they induced goblet cell proliferation in the tracheas of asthmatic mice [47]. Here, we demonstrated that sesamol reduced IL-4 and IL-13 levels in the BALF and lungs. These results suggested that the sesamol inhibition of IL-4 and IL-13 expression caused a reduction in goblet cell proliferation in the trachea.

Previous studies pointed out that, in patients with asthma, the airways are stimulated to induce excessive oxidative stress, which aggravates mucus and sputum production and also damages lung cells [48,49]. Asthmatic mouse models and in vitro cellular experiments showed that allergens or inflammatory cytokines could stimulate airway epithelial cell activation, which increased chemokine release, attracted inflammatory immune cell infiltration into the lungs, and induced ROS production in tracheal epithelial cells [8,50]. Animal experiments also showed that these inflamed immune cells released high levels of inflammatory mediators in the lungs and induced oxidative stress, which damaged lung cells and attenuated lung function [51,52]. In the present study, we found that sesamol inhibited ROS production in inflammatory BEAS-2B cells and reduced CCL11 and CCL24 expression, which inhibited eosinophil infiltration. Moreover, reducing the amounts of eosinophil peroxidase released by eosinophils reduced the oxidative damage to lung cells. Sesamol also reduced the levels of MCP-1and IL-8, which suppressed macrophage (monocyte) and neutrophil migration into the lungs of asthmatic mice. Moreover, high doses of sesamol reduced the numbers of macrophages and neutrophils in BALF. Hence, sesamol suppressed ROS release, and thus maintained lung function in asthmatic mice. We employed a lipid peroxidation assay to detect MDA activity, an identifying marker of oxidative stress in cells or tissues [8]. Additionally, glutathione is an important antioxidant enzyme that regulates oxidative stress in the lungs of patients with asthma [53]. We found that sesamol significantly attenuated the MDA levels and promoted GSH expression in the lungs of asthmatic mice. These activities regulated peroxidation effects in the lungs of asthmatic mice. These findings were consistent with previous studies, which showed that sesamol increased GSH and decreased MDA levels in rat ischemic cortex tissues [24]. Sesamol regulation of GSH and MDA levels also reduced oxidative stress and improved cognitive impairments in mouse brains [54]. Hence, sesamol had a protective effect against oxidative stress, and thus maintained lung function in asthmatic mice. 

## 5. Conclusions

In conclusion, our findings confirmed that sesamol ameliorates airway inflammation, inhibits eosinophil infiltration into the lung, and attenuates tracheal mucus secretion by suppressing Th2-associated cytokine and eotaxin release in asthmatic mice. More specifically, sesamol showed potential for improving inflammation and oxidative stress in asthma.

## Figures and Tables

**Figure 1 antioxidants-09-00295-f001:**
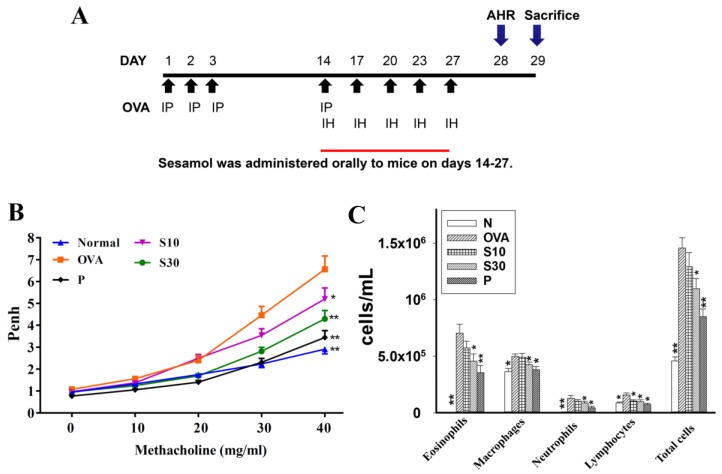
The effect of sesamol on AHR and immune cell counts in the BALF of mice. (**A**) Experimental procedures for asthmatic mouse studies. For sensitization, mice received an intraperitoneal injection (I.P.) of OVA on days 1–3 and 14. Next, mice were challenged with 2% OVA inhalation (I.H.) on days 14, 17, 20, 23, and 27. Different mouse groups were fed saline (normal and OVA groups), sesamol (S10 and S30 groups), or prednisolone (P groups), daily for 2 weeks (days 14-27; n = 8 mice/group). (**B**) Mouse AHRs measured on day 28 with increasing metacholine challenges; results are expressed as Penh values. (**C**) Inflammatory cells measured in BALF. All data are the means ± SEM. * *p* <0.05, ** *p* <0.01 compared to the OVA control group. AHR: airway hyperresponsiveness; BALF: bronchoaveolar lavage fluid; OVA: ovalbumin; Penh: enhanced pause.

**Figure 2 antioxidants-09-00295-f002:**
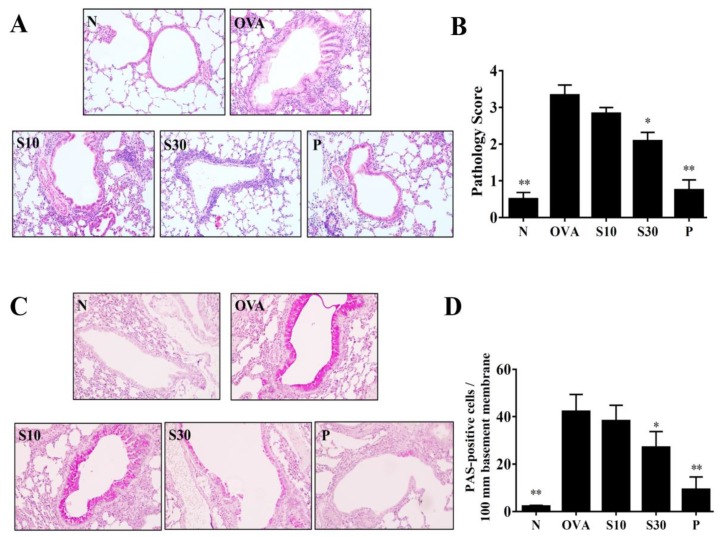
Sesamol effects on asthmatic lung tissues. Histological sections of lung tissues are shown from normal (N) and OVA-stimulated (OVA) mice treated with or without prednisolone (P) or sesamol (S10 and S30). (**A**) HE staining shows eosinophil infiltration (200× magnification); (**B**) pathological scores reflect the degree of eosinophil infiltration in lung sections. (**C**) PAS staining shows goblet cell hyperplasia (200× magnification); (**D**) the number of PAS-positive cells per 100 μm of basement membrane. All data are the means ± SEM. * *p* <0.05, ** *p* <0.01 compared to the OVA control group; HE: hematoxylin and eosin; OVA: ovalbumin; PAS: periodic acid-Schiff.

**Figure 3 antioxidants-09-00295-f003:**
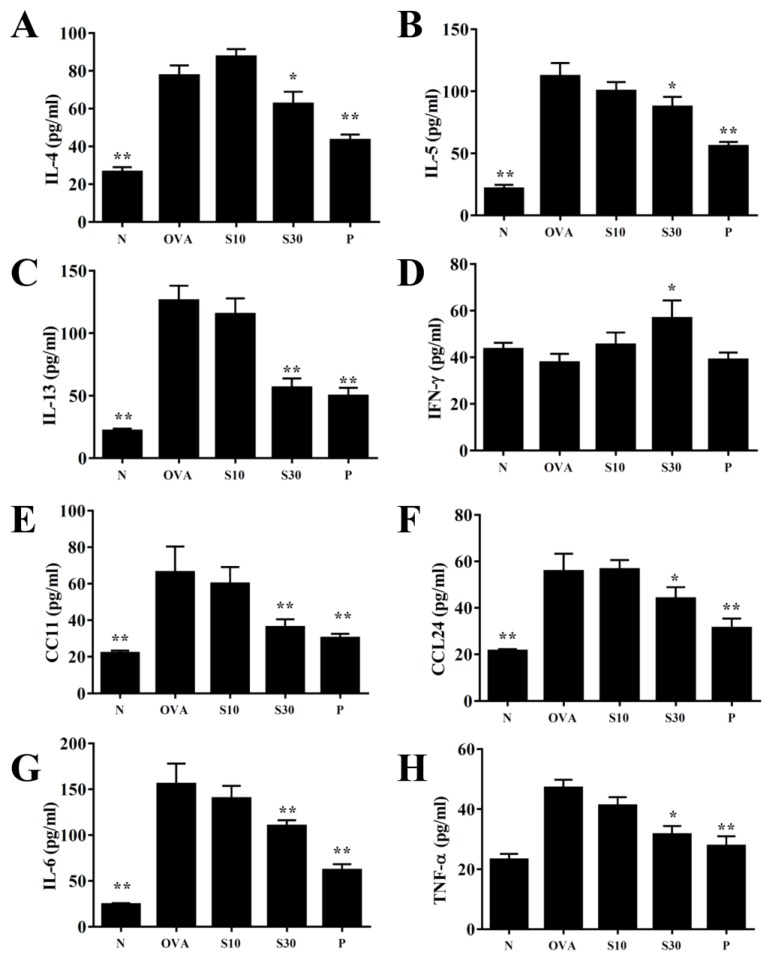
Effects of sesamol on cytokine and chemokine levels in BALF. ELISA results show the concentrations of (**A**) IL-4, (**B**) IL-5, (**C**) IL-13, (**D**) IFN-γ, (**E**) CCL11, (**F**) CCL24, (**G**) IL-6, and (**H**) TNF-α in BALF from normal (N) and OVA-stimulated (OVA) mice treated without or with prednisolone (P) or sesamol (S10 and S30). All data are the means ± SEM. * *p* <0.05, ** *p* <0.01 compared to the OVA control group. BALF: bronchoaveolar lavage fluid; OVA: ovalbumin.

**Figure 4 antioxidants-09-00295-f004:**
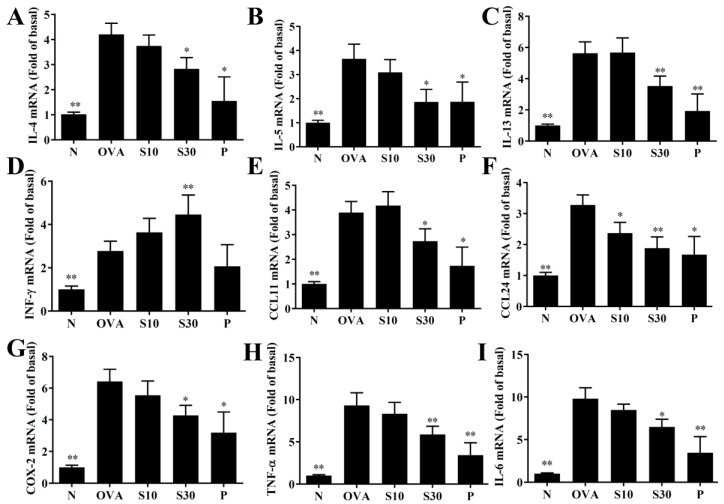
Sesamol effects on gene expression in the lungs. mRNAs extracted from lung tissues were amplified with real-time RT-PCR. Results show gene expression levels in normal (N) and OVA-stimulated (OVA) mice treated without or with prednisolone (P) or sesamol (S10 and S30). The results show the expression levels of (**A**) IL-4, (**B**) IL-5, (**C**) IL-13, (**D**) IFN-γ, (**E**) CCL11, (**F**) CCL24, (**G**) COX-2, (**H**) TNF-α, and (**I**) IL-6. Fold-changes in expression were measured relative to β-actin expression levels (internal control). All data are the means ± SEM. * *p* <0.05, ** *p* <0.01 compared to the OVA control group. OVA: ovalbumin.

**Figure 5 antioxidants-09-00295-f005:**
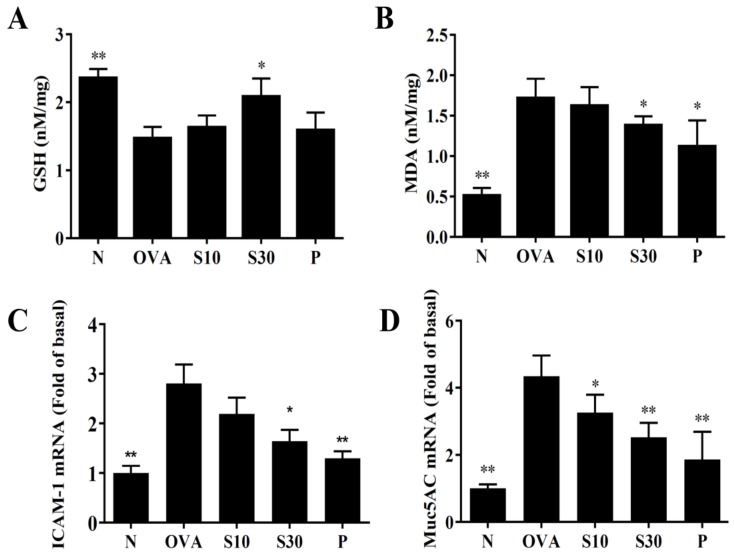
Sesamol regulation of oxidative activities in the lung. (**A**) GSH activity and (**B**) MDA activity were measured in the lung tissues of normal (N) and OVA-stimulated (OVA) mice treated without or with prednisolone (P) or sesamol (S10 and S30). Sesamol suppressed the expression of (**C**) ICAM-1 and (**D**) Muc5Ac genes in the lungs. All data are the means ± SEM. * *p* <0.05, ** *p* <0.01 compared to the OVA control group. ICAM-1: intercellular adhesion molecule 1; GSH: glutathione; MDA: malondialdehyde; musc5Ac: mucin 5Ac protein; OVA: ovalbumin.

**Figure 6 antioxidants-09-00295-f006:**
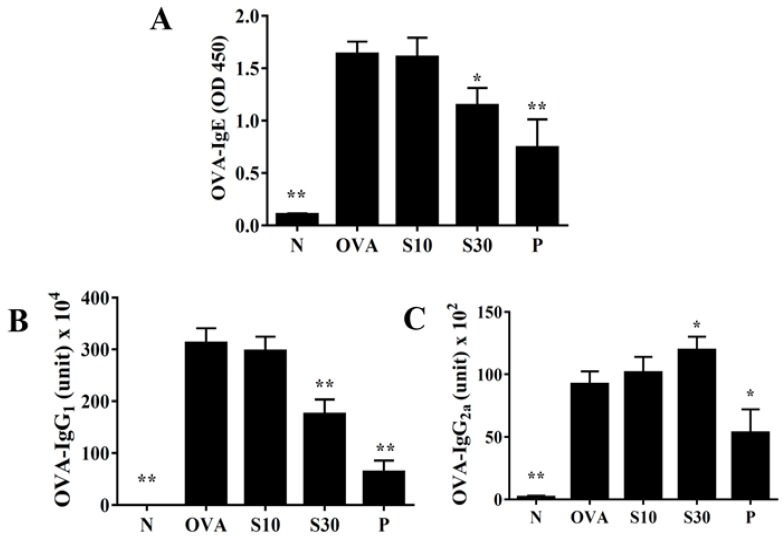
Sesamol effects on OVA-specific antibodies in serum. Serum levels of (**A**) OVA-IgE, (**B**) OVA-IgG_1_, and (**C**) OVA-IgG_2a_ are shown for normal (N) and OVA-stimulated (OVA) mice treated without or with prednisolone (P) or sesamol (S10 and S30). All data are the means ± SEM. * *p* <0.05, ** *p* <0.01 compared to the OVA control group. OVA: ovalbumin.

**Figure 7 antioxidants-09-00295-f007:**
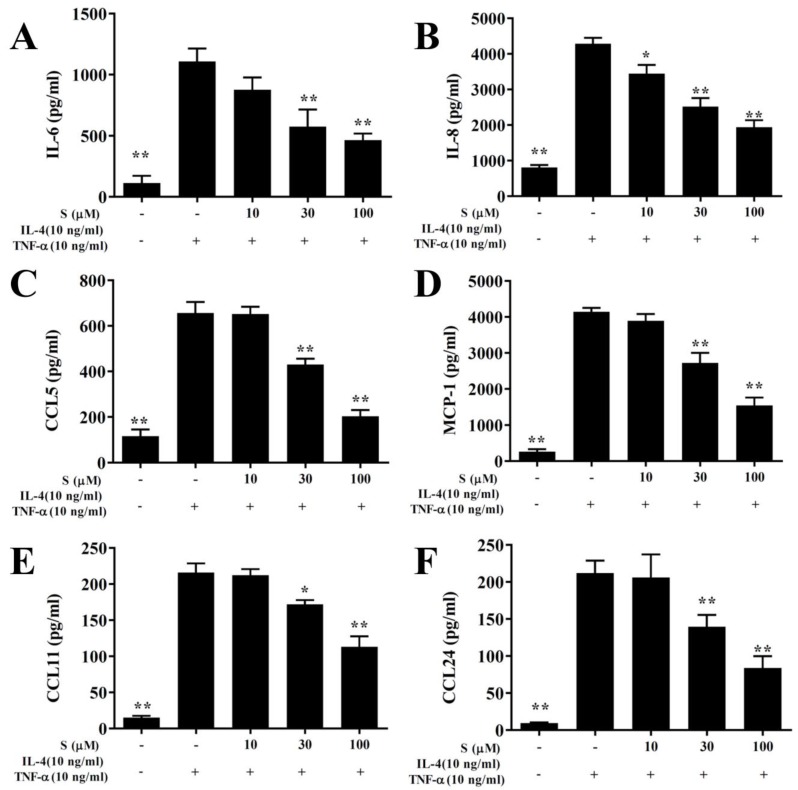
Sesamol effects on cytokine and chemokine production in BEAS-2B cells. ELISA results show sesamol (S) effects on the concentrations of (**A**) IL-6, (**B**) IL-8, (**C**) CCL5, (**D**) MCP-1, (**E**) CCL11, (**F**), and CCL24 in TNF-α/IL-4-activated BEAS-2B cells. The data represent the mean ± SEM; * *p* <0.05, ** *p* <0.01, compared to BEAS-2B cells stimulated with 10 ng/mL TNF-α/IL-4.

**Figure 8 antioxidants-09-00295-f008:**
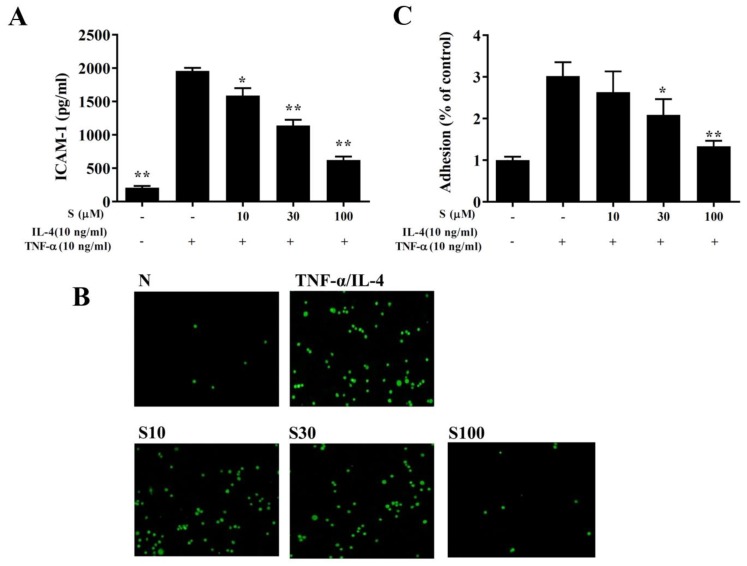
Sesamol inhibited THP-1 cell adherence to BEAS-2B cells. (**A**) Sesamol (S) reduced the levels of ICAM-1 in TNF-α/IL-4-activated BEAS-2B cells. (**B**) Fluorescence images show the adhesion of THP-1 cells to TNF-α/IL-4-stimulated BEAS-2B cells. (**C**) Fluorescence intensities were evaluated to determine the percent of THP-1 adhesion to BEAS-2B cells. The data represent the mean ± SEM of three independent experiments; * *p* <0.05, ** *p* <0.01, compared to BEAS-2B cells stimulated with TNF-α and IL-4.

**Figure 9 antioxidants-09-00295-f009:**
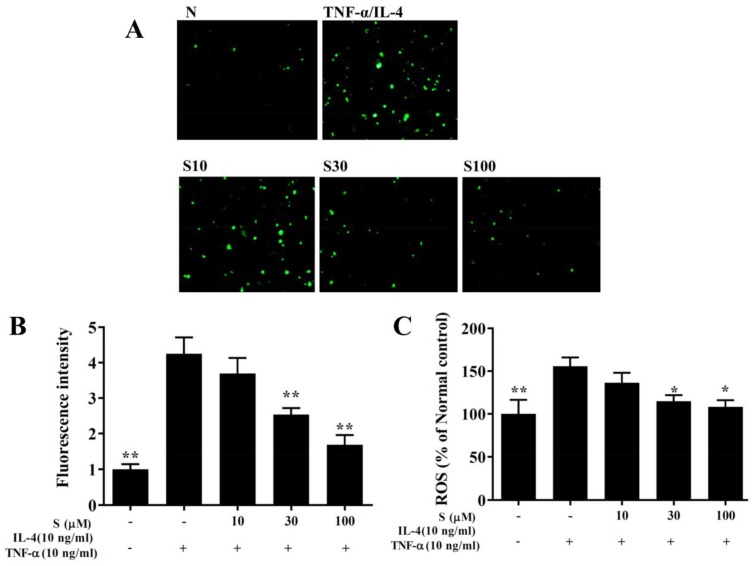
Sesamol effects on ROS production in activated BEAS-2B cells. (**A**) Fluorescence images show intracellular ROS in normal inactivated (N) and in TNF-α/IL-4-activated BEAS-2B cells treated without or with sesamol (S). (**B**) Fluorescence intensities indicate the levels of intracellular ROS in BEAS-2B cells. (**C**) The percentage of ROS detected in TNF-α/IL-4-activated BEAS-2B cells compared to that detected in normal control cells. All data represent the mean ± SEM; * *p* <0.05, ** *p* <0.01, compared to BEAS-2B cells stimulated with TNF-α and IL-4.

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
