# Peer review of "Sesamol Alleviates Airway Hyperresponsiveness and Oxidative Stress in Asthmatic Mice"

_antioxidants, 2020, doi:10.3390/antiox9040295_

Round 1

Reviewer 1 Report

I read with interest the manuscript of Liou et al. The study is well-designed and thoroughly examines the effect of sesamol on eosinophilic airway inflammation. 

Comments:

  1. What are the side effects of sesamol treatment on animals? E.g. weight loss, gain, death
  2. animals were fed with sesamol at a concentration of 10 and 30 mg/kg. Translating it to the human situation how much sesame oil should be taken daily by asthma patients to expect some anti-inflammatory effects?
  3. the ova model is a classical model, however house dust mite sensitization resembles the human situation better. 
  4. what intracellular pathways are expected to be modulated by the sesamole to suppress th2 inflammation?  

Author Response

I read with interest the manuscript of Liou et al. The study is well-designed and thoroughly examines the effect of sesamol on eosinophilic airway inflammation.

Comments:

1.What are the side effects of sesamol treatment on animals? E.g. weight loss, gain, death

Responses:

Thank for reviewer’s suggestion. We have investigated that C57BL/6 mice were fed a high-fat diet (HFD) to induce obesity and non-alcoholic fatty liver disease, and then oral administration with sesamol. Our results demonstrated that HFD-induced obese mice treated with sesamol had decreased body weight as well as inguinal and epididymal adipose tissue weights compared with HFD-treated mice. However, mice were fed normal chow diet and fed with sesamol did not increase or decrease body weight compared to mice were fed normal chow diet (Experimental results unpublished). Furthermore, we also assayed ALT and AST levels in serum, and found that obese mice fed with sesamol could decrease ALT and AST levels in treated mice compared with the HFD-induce obese mice (Experimental results unpublished). Hence, we thought that sesamol has no side effects in animal experiment.

2.animals were fed with sesamol at a concentration of 10 and 30 mg/kg. Translating it to the human situation how much sesame oil should be taken daily by asthma patients to expect some anti-inflammatory effects?

Responses:

(50 kg) of people need to take 0.5-1.5g of sesamol orally every day. We hope that sesamol (not sesame oil) can be made into capsules as a healthy food to improve the symptoms of asthma.

3.the ova model is a classical model, however house dust mite sensitization resembles the human situation better.

Responses:

OVA, house dust mite, pollen, and fungi could induce asthma-like pathology in mice. House mite dust extract can be purchased from Greer Laboratories Inc. But there is no Greer Laboratories Inc agent in Taiwan. House dust mite extract or mite allergen Derp2 is not easy to buy in Taiwan. We had purified Derp2 from transgenic yeast, but the result did not significantly induce asthma-like pathology. Hence, we most commonly used OVA to induce asthma-like pathology in mice.

4.what intracellular pathways are expected to be modulated by the sesamole to suppress th2 inflammation?

Responses:

Our earlier studies demonstrated that sesamol significantly inhibited expression of proinflammatory cytokines and mediators by suppressing the activation of NF-κB and MAPK pathways and promoting AMPK activation. Sesamol also displayed an antioxidant effect by activating the Nrf2/HO-1 pathway in LPS-activated murine

macrophages. In this study, sesamol significantly reduced proinflammatory cytokine levels and eotaxin in inflammatory BEAS-2B cells. Hence, we thought that sesamol maybe decrease inflammatory response of inflammatory BEAS-2B cells and through blocked NF-κB and MAPK signal pathways.

Reviewer 2 Report

Chian-Jiun Liou et al. presented data on Sesamol, a polyphenol lignin, able to reduce airway hyperresponsivness and oxidative stress in a mouse model of asthma and in tracheal epithelial cells. They found that oral sesamol administration significantly suppressed eosinophil infiltration in the lung, airway hyperresponsivness and Th2 cytokine expression. Sesamol significantly increased glutathione and decreased MDA in lung of asthmatic mice. Furthermore, Sesamol reduced ROS and ICAM expression. Data are clearly presented and support conclusions.

COMMENTS:

  1. Sesamol could induce apoptosis in cancer cells and inhibit the effects of proliferation and autophagy in lung cancer and colon cancer. Did the authors evaluate the possible toxic or cytostatic effect of sesamol in tracheal epithelial cells? If evaluated in previous works, a short sentence with bibliographic reference should be added.
  2. The authors should add when blood was taken to evaluate OVA serum IgG and IgE levels. Did sesamol affect total IgG or IgE levels?
  3. Figure 1C: When local inflammatory cells are evaluated, macrophages usually are considered instead of monocytes which are blood leukocytes. All cell populations should be plural names.
  4. Is the decrease of lung eosinophils associated with a decrease in blood eosinophils in sesamol treated mice?
  5. Results, pg 5, line 201-202: “eosinophils in OVA-treated asthmatic mice were significantly reduced with sesamol”, please add “with high sesamol concentration”.
  6. Results, pg 6, line 221 and Figure 2B: please define how “pathology score” was calculated.
  7. Results are always shown as means and SEM. Are data normally distributed? If yes, please add the evaluation in the statistical method section, otherwise show data as medians and ranges.
  8. Did the authors evaluate if sesamol can synergistically act with prednisolone to reduce IL-5 or other cytokines/chemokines?
  9. The authors should add how they scored fluorescence intensity in figure 9.
  10. Higher concentrations of Sesamol increased IFNg production in BALF and mRNA expression in BEAS-2B cells. The authors commented in the discussion that “Th2 cytokines inhibit the expression of Th1 cells which reduce IFNg production”. The authors should comment the that sesamol increased IFNg more than prednisolone in the experimental conditions considered.
  11. Discussion: pg 12, line 363: Airway remodeling is not “an important pathological symptom” but a structural change of the airways of asthmatic patients, which can determine asthma symptoms.
  12. Did the authors have any data regarding a possible effect of sesamol during the sensitization phase of OVA-asthma model?
  13. Most of the bibliographic references are reviews. The authors should, when possible, substitute them with the original studies.

Author Response

Chian-Jiun Liou et al. presented data on Sesamol, a polyphenol lignin, able to reduce airway hyperresponsivness and oxidative stress in a mouse model of asthma and in tracheal epithelial cells. They found that oral sesamol administration significantly suppressed eosinophil infiltration in the lung, airway hyperresponsivness and Th2 cytokine expression. Sesamol significantly increased glutathione and decreased MDA in lung of asthmatic mice. Furthermore, Sesamol reduced ROS and ICAM expression. Data are clearly presented and support conclusions.

COMMENTS:

1.Sesamol could induce apoptosis in cancer cells and inhibit the effects of proliferation and autophagy in lung cancer and colon cancer. Did the authors evaluate the possible toxic or cytostatic effect of sesamol in tracheal epithelial cells? If evaluated in previous works, a short sentence with bibliographic reference should be added.

Responses:

Our earlier studies demonstrated that sesamol did not cause significant cell cytotoxicity at doses ≤100 μM in macrophage RAW 264.7 cells. In this study,

the cytotoxicity of sesamol in BEAS-2B cells was determined using the MTT assay.

Sesamol did not show significant cytotoxic effects at doses ≤100 μM, and subsequent experiments used sesamol at 10–100 μM concentrations. We would describe the results of cell viability in this manuscript. pg 8, line 271-273

2.The authors should add when blood was taken to evaluate OVA serum IgG and IgE levels. Did sesamol affect total IgG or IgE levels?

Responses:

Serum were collect to detect OVA-IgE and total IgE levels on days 14, 20, and 29. We found that sesamol did not significantly modulate the serum levels of OVA-IgE and total IgE, compared to OVA-treated mice on days 14, and 20. However, sesamol could significantly decrease the serum levels of total IgE, compared to OVA-treated mice on days 29. Additionally, sesamol did not also regulate the serum levels of total IgG compared to OVA-treated mice on days 14, 20, and 29. In this study, we want to emphasize that OVA-sensitized mice can produce OVA-specific antibodies. Hence, we presented the levels of OVA-IgE, OVA-IgG1, and OVA-IgG2a in this manuscript.

3.Figure 1C: When local inflammatory cells are evaluated, macrophages usually are considered instead of monocytes which are blood leukocytes. All cell populations should be plural names.

Responses:

Thank for reviewer’s suggestion. We modified those mistakes in this manuscript.

4.Is the decrease of lung eosinophils associated with a decrease in blood eosinophils in sesamol treated mice?

Responses:

Our experimental results did not confirm that decreased of lung eosinophils associated with a decrease in blood eosinophils in sesamol treated asthma mice.

We did not measure the eosinophil numbers in the blood, but we detected the levels of CCL11 and CCL24 (eosinophil-associated chemokines) in the serum. We found that sesamol did not reduce the serum levels of CCL11 and CCL24 compared to OVA-treated mice. However, in BALF and lung, sesamol significantly reduced the levels of CCL11 and CCL24, compared to OVA-sensitized asthmatic mice. We hypothesized that sesamol could improve local asthma symptoms in the lungs of asthma mice.

5.Results, pg 5, line 201-202: “eosinophils in OVA-treated asthmatic mice were significantly reduced with sesamol”, please add “with high sesamol concentration”.

Responses:

Thank for reviewer’s suggestion. We modified the sentence in this manuscript. pg 5, line 209-210

6.Results, pg 6, line 221 and Figure 2B: please define how “pathology score” was calculated.

Responses:

We described the pathological scores (eosinophil infiltration) in the section “2.4 Histological analysis of lung tissue”. pg 4, line 118-123

7.Results are always shown as means and SEM. Are data normally distributed? If yes, please add the evaluation in the statistical method section, otherwise show data as medians and ranges.

Responses:

Thank for reviewer’s suggestion. We modified the sentence about “2.14. Statistical analysis” in this manuscript. Statistical analyses were performed with a one-way analysis of variance (ANOVA), followed by Dunnett’s post hoc test for normally distributed data. Non-normally distributed data would use non-parametric Kruskal-Wallis analysis. All data are expressed as the means ± standard error of the mean (SEM), based on at least three independent experiments. P-values <0.05 were considered significant.

8.Did the authors evaluate if sesamol can synergistically act with prednisolone to reduce IL-5 or other cytokines/chemokines?

Responses:

Thank for reviewer’s suggestion. In our experiments, we did not design experiments on the combined sesamol with prednisolone to treat asthmatic mice. Hence, we can't confirm if there is a synergistic effect between sesamol and prednisolone to improve asthma symptom. Reviewer’s command is interesting. We hope that supplementation with sesamol can reduce the prednisolone dosage to reduce asthma symptoms or regulated the levels of Th2 cells-associated cytokines and chemokines, which would increase the value of sesaminol as a healthy food. We will design an experiment that combined sesaminol with different doses of prednisolone (or Corticosteroids) to evaluate airway hyperresponsiveness, eosinophil infiltration, and the levels of cytokine and chemokine in asthmatic mice.

9.The authors should add how they scored fluorescence intensity in figure 9.

Responses:

Three images are calculated for each result, and three areas are selected for each image to quantify the fluorescence intensity. We would describe the fluorescence quantitative method in the section “2.13. Reactive oxygen species assay”. pg 5, line 187-189

10.Higher concentrations of Sesamol increased IFNg production in BALF and mRNA expression in BEAS-2B cells. The authors commented in the discussion that “Th2 cytokines inhibit the expression of Th1 cells which reduce IFNg production”. The authors should comment the that sesamol increased IFNg more than prednisolone in the experimental conditions considered.

Responses:

Responses:

Thank for reviewer’s suggestion. We modified the sentence in this manuscript. pg 12, line 350-356

11.Discussion: pg 12, line 363: Airway remodeling is not “an important pathological symptom” but a structural change of the airways of asthmatic patients, which can determine asthma symptoms.

Responses:                                                                                                                       

Thank for reviewer’s suggestion. We modified those mistakes in this manuscript. pg 13, line 378

12.Did the authors have any data regarding a possible effect of sesamol during the sensitization phase of OVA-asthma model?

Responses:

Serum IgE concentration is an important indicator for detecting asthma. Hence, mice were collect serum to detect OVA-IgE levels on days 14, 20, and 29. We found that sesamol did not significantly modulate the serum levels of OVA-IgE, compared to OVA-treated mice on days 14, and 20. However, sesamol could significantly decrease the serum levels of OVA-IgE, compared to OVA-treated mice on days 29. Hence, we presented the levels of OVA-IgE, OVA-IgG1, and OVA-IgG2a in this manuscript.

13.Most of the bibliographic references are reviews. The authors should, when possible, substitute them with the original studies.

Responses:

We modified some references in this manuscript.

Round 2

Reviewer 2 Report

The authors properly answered to the comments.